# The Variables of the Readiness for Discharge from Hospital in Patients after Myocardial Infarction

**DOI:** 10.3390/ijerph20021582

**Published:** 2023-01-15

**Authors:** Ewelina Kolarczyk, Agnieszka Witkowska, Marek Szymiczek, Agnieszka Młynarska

**Affiliations:** 1Department of Gerontology and Geriatric Nursing, Faculty of Health Sciences, Medical University of Silesia, 40-635 Katowice, Poland; 2Department of Cardiology, Electrotherapy and Angiology, Scanmed S.A. Racibórz Medical Center, 47-400 Racibórz, Poland

**Keywords:** myocardial infarction, chronic illness, discharge from hospital

## Abstract

Discharge after myocardial infarction (MI) reduces the risk of repeated myocardial infarction and stroke and has a positive effect on the patient’s prognosis. An important element of preparation is the assessment of the patient’s readiness for discharge from hospital. This study aimed to evaluate the associations between a patient’s readiness for hospital discharge after MI, their functioning in the chronic illness, and socio-demographic and clinical variables. Methods: This was a cross-sectional, single-center study. The study was conducted among 242 patients who were hospitalized for myocardial infarction after percutaneous coronary intervention (PCI). The Readiness for Hospital Discharge After Myocardial Infarction Scale (RHD-MIS) and the Functioning in Chronic Illness Scale (FCIS) were used. Results: No statistically significant differences were found between socio-demographic and clinical factors and the overall result of the RHD-MIS (*p* >0.05).There is a positive correlation between hospital discharge readiness and functioning in chronic disease in patients after MI (r = 0.20; *p* < 0.001). The higher the level of subjective knowledge, the better the functioning in chronic disease (rho = 0.16; *p* < 0.05), the greater the increase in the sense of influence on the course of the disease (rho = 0.17; *p* < 0.05) and the greater the decrease in the impact of the disease on the patient’s attitude (rho = 0.23, *p* < 0.05). Conclusions: The higher the readiness for discharge from hospital, the better the patient’s functioning in the disease and the lower the impact of the disease on the patient.

## 1. Introduction

The concept of “hospital discharge readiness” was first defined in 1976 by Fenwick and referred to the patient’s sense of readiness to face reality [1,2]. Discharge readiness includes the physical stability of the patient about to be discharged home, and whether they have adequate support, psychological capacity, and information and knowledge [2]. According to Galvin et al., physical stability refers to physical self-care, and self-care and psychological stability refers to the patients’ subjective feeling regarding their ability to cope after returning home. Knowledge refers to the patient’s information about myocardial infarction, as well as the scope of self-care in this disease entity. “Support” isunderstood as the emotional and psychological support received by the patient from their family, medical facility, or social organization [2]. According to Mabire et al. (2019), patient readiness for hospital discharge is influenced by the discharge process and structural factors such as the patient population, ward size and educational experience of nurses in the team. It is believed that patient readiness to be discharged from hospital is lower in larger medical units [3].A patient with a history of myocardial infarction is at high risk of subsequent cardiovascular events, such as another myocardial infarction, stroke and death. Teaching patients correct health behaviors after myocardial infarction contributes to a reduced risk of the recurrence of myocardial infarction and has a positive effect on the prognosis of such a patient [4]. It is strongly recommended that the use of hospital discharge readiness assessments becomes a fundamental part of the routine process of patients’ hospital discharge and return to the home environment [5]. It is believed that the importance of readiness for hospital discharge is an issue that should be analyzed worldwide and related to the search for ways to reduce healthcare costs [1,6]. The effect of properly preparing patients for discharge from hospital is in obtaining better results in terms of self-care, thus reducing the number of rehospitalizations, visits to emergency departments and even deaths [7]. Previous scientific studies confirm that there is arelationship between the assessment of readiness for hospital discharge and adherence to treatment in long-term follow-up in patients after myocardial infarction [8] and acceptance of the disease in patients after myocardial infarction [9].As reported by other studies, there is arelationship between the patient’s educational expectations and their long-term adherence to treatment [10]. Although there have been previous studies on readiness for hospital discharge, knowledge about readiness for discharge in patients with a history of myocardial infarction remains limited.

This study aimed to examine the level of hospital discharge readiness reported by patients in relation to the level of functioning in chronic illness and its socio-demographic factors among patients with myocardial infarction. The main research problems are presented in the following questions: (1) whether and how the level of functioning of patients with chronic illness after myocardial infarction affects these patients’ readiness to be discharged from hospital; (2) whether and which socio-demographic factors affect the readiness of patients to be discharged from the hospital after myocardial infarction; and (3) whether and which socio-demographic factors affect the level of functioning in chronic illness in patients after myocardial infarction.

## 2. Materials and Methods

### 2.1. Study Design

A cross-sectional, single-center study was carried out. The study was carried out according to the STROBE checklist for observational studies [11].

### 2.2. Settings, and Participants

This study used a selected group of patients (*n* =242) in the Unit of Cardiology, Electrotherapy and Angiology Scanmed S.A. Racibórz Medical Center, in the Silesian Voivodeship. The patients were hospitalized due to myocardial infraction within the period between June 2021 and June 2022. The written consent of the management of Scanmed S.A. was obtained for the study. The minimum sample size was 196, which was calculated based on the available patient population, with a 95% confidence interval. The data used to determine the minimum number of individuals within the group were obtained from the demographic situation in Poland up to 2020 [12].The questionnaires were provided in a paper form (“pen-pencil questionnaire”) and the interview was conducted by two investigators who were medical staff on this ward and one investigator who was a researcher from university. The surveys were collected on the last day of hospitalization. The inclusion criteria for the study wereadult participants: (1) diagnosed with myocardial infarction, as well STEMI as NSTEMI; (2) treated with percutaneous coronary intervention (PCI); (3) who provided informed consent to participate in the study; (4) with no dementia-related disorders; (5) with no mental disorders. Patients who did not consent to participate in the study, or who were unable to answer questions due to hearing disorders or vision disorders, advanced senile dementia or diagnosed mental illnesses were excluded from the study.

### 2.3. Instruments

The research was carried out using two standardized measures: the Readiness forHospital Discharge After Myocardial Infarction Scale (RHD-MIS) and the Functioning in Chronic Illness Scale (FCIS). Moreover, an original 14-question survey on socio-demographic and clinical factors was used.

The RHD-MIS scale evaluates patients’ readiness for discharge after myocardial infarction. The questionnaire included three subscales: (1) subjective assessment of patient knowledge about the disease, (2) objective assessment of patient knowledge about the disease, (3) patient expectations. To measure readiness, the centile standards were constructed to express low, intermediate and high values. A score from 0 to 3 was assigned to each RHD-MIS item. A patient who scores above 57 points has high readiness, patients with scores less than 44 points have low readiness, while a score ranging from 44 to 57 points indicates a patient with an intermediate level of readiness. RHD-MIS is considered a reliable and relevant tool for measuring patient readiness for discharge. The a-Cronbach coefficient was 0.789, indicating a high level of reliability and homogeneity [13].

The Chronic Disease Functioning Scale (FCIS) obtains results in four scales: general functioning of the patient in the illness (scores in the range of 24–120 points, where scores up to 78 points mean a low level, scores of 79–93 mean a medium level and scores above 93 points mean a high level), the impact of the illness on the patient (8–40, where the low level is up to 23, the medium level is 24–33 and the high level is above 33), the impact of the patient on the illness (8–40, where the low level is up to 24, the level average 25–29 and high level above 29) and the impact of the illness on the patient’s attitudes (8–40, where low level is up to 27, medium level is 28–33 and high level is above 33).The internal consistency of the questionnaire expressed by the a-Cronbach coefficient was 0.855, which indicates its high reliability and homogeneity. A set of items divided into three subscalesallows for the evaluation of the impact of the disease on the patient, the patient’s impact on the disease and the impact of the disease on the patient’s attitudes [14].

### 2.4. Statistical Analyses

The analysis used the significance level *p* < 0.05.Parametric tests (Student’s *t*-test or ANOVA analysis of variance) or their non-parametric equivalents (Mann–Whitney U test or Kruskal–Wallis test) were used to analyze quantitative variables when broken down into groups. The Bonferroni’s post hoc test was performed to determine precisely which differences were significant between groups. The correlation between variables was verified using the Pearson’s (*r*) or Spearman’s (*rho*) correlation coefficient. Tests were selected on the basis of the distribution of variables, which was verified by the Shapiro–Wilk test.The mean (M), standard deviation (SD), median (Me), first quartile (Q1) and third quartile (Q3) were analyzed. Calculations were carried out in the statistical environment R ver.3.6.0, PSPP program and MS Office 2019.

### 2.5. Ethical Procedure

The study was conducted under the recommendations of the Helsinki Declaration implemented by the World Medical Association [15] and the guidelines of Good Clinical Practice [16]. Before beginning the study, the respondents were informed about the anonymous and voluntary nature of the survey. Consent to participate in the study was obtained from each respondent. The study protocol was approved by the Bioethics Committee at the Medical University of Silesia in Katowice on 04 March 2019 (ethical approval code: KNW/0022/KB/46/19).

## 3. Results

### 3.1. Characteristics of the Study Group

The sample consisted of 242 patients (80 women and 162 men). The ages of the respondents were ≥70 years(72(29.9%)), 70–61 years(83(34.9%)), 60–51 years (53(22%)), 50–41 years (22(9.1%)) and ≤ 40 years (10(4.1%)). The majority of respondents were married (162;71.2). Most of the surveyed patients were city dwellers (123;59.9%) and had children (205;92,4%). The majority of the respondents were pensioners (103;54.2%), nearly 6.8%(13) were unemployed and 74 (38.9%) of the respondents were working people. The epidemiological information about the prevalence of chronic diseases indicated the coexistence of hypertension (159;77.7%), diabetes (73;35.4%), lipid disorders (58;28.2%) and other diseases (74;36.9%). Most of the surveyed patients had had one episode of myocardial infarction (199;82.2%), nearly 15.7%(37) had had two episodes and 52.1%(5) had had more than two episodes of myocardial infarction.

### 3.2. Readiness for Hospital Discharge (RHD-MIS)

Of the 242 respondents, 217 questionnaires that contained complete answers to the questions were included in the analysis of the hospital discharge readiness survey data. The study indicated low readiness for hospital discharge in 89(40.6%) patients, while 94(42.9%) showed an intermediate level and 36(16.5%) exhibited high readiness for discharge. The analysis showed no statistically significant differences between socio-demographic and clinical factors, and the overall results of the hospital discharge readiness scale in patients after myocardial infarction (*p* > 0.05). Therefore, the assumed hypothesis that socio-demographic and clinical variables significantly differentiate the degree of hospital discharge readiness in patients after myocardial infarction had to be rejected. The results are presented in Table 1.

#### 3.2.1. RHD-MIS Subscale 1: The Assessment of Subjective Knowledge

Regarding the RHD-MIS subscales, we observed statistically insignificant (*p* > 0.05) differences in the level of subjective knowledge about socio-demographic and clinical variables. The study shows that subjective knowledge was slightly higher among men, people up to 60 years of age, overweight people, unmarried people, those living in the countryside, those with secondary education and professionally active people. Slightly more subjective knowledge was also observed among people with children, those without comorbidities, those not smoking at all and people after their first heart attack (*p* > 0.05). The results are presented in Table 2.

#### 3.2.2. RHD-MIS Subscale 3: The Assessment of Objective Knowledge

The Mann–Whitney U test showed that respondents’ objective knowledge showed statistically significant (*p* < 0.05) differences in terms of objective knowledge between people with and without children, as well as differences between people suffering from other chronic diseases and people without such diseases.

Over half of the respondents with children (184;16.39 ± 4.64), interms of objective knowledge, obtained results no lower than *Me* = 18.00 (the other half of those with children obtained a result no higher than *Me* = 18.00). The lowest score among this group was Min = 1.00 and the highest was Max = 21.00. Half of those without children obtained a score no higher than *Me* = 12.00 (the other half obtained a score no lower than *Me* = 12.00). The lowest score was Min = 8.00; the highest was Max = 21.00. People with children had significantly (*p*= 0.033) greater objective knowledge than people without children. Half of the people without other chronic diseases had a score no higher than *Me* = 17.00. The lowest score among this group was Min = 1.00, and the highest was Max = 21.00. Half of those with other chronic diseases had a score no lower than *Me* = 19.00. The lowest score was Min = 6.00; the highest was Max = 21.00. People with other chronic diseases were characterized by statistically significantly (*p* = 0.001) greater objective knowledge than people without such diseases. The distribution of variables is shown in Figure 1.

#### 3.2.3. RHD-MIS Subscale 3: The Assessment of Expectations

Statistical analysis showed statistically significant (*p* = 0.023) differences in the level of expectations between men and women. Half of the women obtained a score no lower than *Me* = 15.00. The lowest score among this group was Min = 0.00, and the highest was Max = 27.00. Half of the men scored no more than *Me* = 10.00. The lowest score was Min = 0.00; the highest was Max = 27.00. Women obtained statistically significantly (*p* < 0.05) higher scores on the expectations scale, and thus felt significantly less need to obtain additional information about their disease and its treatment. The distribution of variables is shown in Figure 2.

### 3.3. The Functioning in Chronic Illness Scale (FCIS)

This study indicated low levels of disease functioning in 76 (31.4%) patients, while118 (48.8%) were characterized by intermediate levels and 48 (19.8%) exhibited high general disease functioning. The U Mann–Whitney test showedstatistically significant (*p* < 0.05) differences in the general functioning of the disease, depending on their professional activity (*p* = 0.026), hypertension (*p* = 0.034), diabetes (*p* < 0.001) and whether they had other chronic diseases (*p* = 0.040). Professionally active people, *M* = 86.04 (*SD* = 11.63) and people without other chronic diseases, *M* = 84.16 (*SD* = 10.99) were characterized by statistically significantly (*p* < 0.05) better overall disease functioning than professionally inactive people, *M* = 82.03 (*SD* = 12.18) and those with other chronic diseases, M = 80.43 (*SD* = 13.60). Half of the respondents without hypertension or diabetes had a result no lower than *Me* = 86.00. Half of the people with hypertension had a result no higher than *Me* = 82.00, and those with diabetes scored no higher than *Me* = 78.00. Thus, it was shown that people suffering from hypertension and diabetes had statistically significantly worse disease functioning than healthy people. The results are presented in Table 3.

#### 3.3.1. Part 1 of FCIS Scale: The Impact of Illness on Patient

The *t*-test for independent trials showed statistically significant (*p* < 0.05) differences in the illness’ level of impact on the patient depending on the presence of diabetes (*p* = 0.007) and other chronic diseases (*p* = 0.043). The results in terms of disease impact among people without diabetes M = 27.23 (SD = 5.36) and other diseases M = 27.13 (SD = 4.97) were statistically significantly (*p* < 0.05) higher than those for people suffering from diabetes M = 25.12 (SD = 5.74) or other chronic diseases M = 25.38 (SD = 6.56). This means that, among people who did not have diabetes or other chronic diseases, the disease had a significantly lower impact on functioning in the chronic illness. The results are presented in Figure 3.

#### 3.3.2. Part 2 of FCIS Scale: The Patient’s Impact on the Illness

The correlation analysis showed that there are statistically significant (*p* < 0.05) differences in the illness’ impact on patients depending on gender, age, professional activity and the presence of hypertension and diabetes. A post hoc Bonferroni test showed significant statistical differences (*p* < 0.05) between people aged over 70 and people in both of the younger age groups. Half of people aged over 70 belived that they have an impact of illness no greater than *Me* = 25.50, and half of the people aged up to 60 and between 61 and 70 believed the impact was no less than *Me* = 28.00. People over 70 years of age found that they had a statistically significantly lower impact than younger people. Professionally active people had a statistically significantly (*p* < 0.05) greater sense of the illness’ impact, *M* = 28.70 (*SD* = 3.75), than professionally inactive people, *M* = 27.30 (*SD* =4.19). Detailed data are presented in Table 4.

#### 3.3.3. Part 3 of FCIS Scale: The Impact of Illness on Patient’s Attitude

The analysis showed that professionally active people (*Me* = 31.00), people with children (*Me* = 30.00) and people without diabetes (*Me* = 30.00) found that the illness had a statistically significantly (*p* < 0.05) lower impact on their attitudes; thus, they had a more optimistic outlook on the future than professionally inactive people (*Me* = 29.00), people without children (*Me* = 27.00) and people suffering from diabetes (*Me* = 27.00). The results of the impact of illness on the patient’s attitude are presented in Table 5.

### 3.4. Correlation between Readiness for Discharge from Hospital after MI and Functioning with Chronic Illness

Pearson’s correlation coefficient analysis showed a statistically significant (*p* < 0.05) positive correlation (r/rho ≤ 0.3) between general readiness for hospital discharge and general functioning in chronic disease (r = 0.20; *p* < 0.01), as well as the disease’s impact on attitudes (r = 0.21; *p* < 0.01). Spearman’s correlation coefficient analysis showed a significant correlation between the level of subjective knowledge and disease functioning (rho = 0.16; *p* < 0.05), the disease’s impact on patients (rho = 0.17; *p* < 0.05) and the disease’s impact on attitudes (rho = 0.23; *p* < 0.01). There was a significant correlation between the level of objective knowledge and general functioning in the disease (rho = 0.15; *p* < 0.05) and the disease’s impact on attitudes (rho = 0.17; *p* < 0.05). There was also a significant positive and weak correlation between the level of patient expectations and the disease’s impact on the patient (rho = 0.15; *p* < 0.05).

## 4. Discussion

This study examined the relationship between readiness for hospital discharge in patients after MI, socio-demographic and clinical variables, and the functioning with chronic illness. The study showed that socio-demographic and clinical variables do not differentiate the general level of hospital discharge readiness or the level of subjective knowledge and expectations in terms of a patient’s readiness (except for higher expectations in the group of women). Objective knowledge regarding hospital discharge readiness correlated positively with having children (*p* = 0.033) and comorbidities (*p* = 0.001).

This study indicated a low readiness for hospital discharge in 89(40.6%) patients, while 94(42.9%) were characterized by an intermediate level and 36(16.5%) exhibited high readiness for discharge. Similar results were obtained in a study conducted by Hydzik et al. in a cardiac rehabilitation ward among 102 patients after myocardial infarction. The cited study showed low readiness in 47.06% of patients, while 27.45% were characterized by intermediate levels and 25.49% exhibited high readiness for discharge [12]. Although the results of the empirical research were essentially consistent with what is presented in the literature on the subject, several separate, interesting observations were made. In the study by Hydzik et al., a higher level of readiness forhospital discharge was found in patients in a relationship, those living in a city, professionally active patients, those without diabetes and those suffering from lipid disorders [9]. Our analysis showed no statistically significant differences between socio-demographic and clinical factors and the overall results of the hospital discharge readiness scale in patients after myocardial infarction (*p* > 0.05). Our results are consistent with those of Kosobucka et al., in a study conducted on 213 patients According to multiple comparison tests, none of the analyzed socio-demographic or clinical factors were associated with the general score of readiness for hospital discharge [8]. Our research, together with the studies cited above, shows that hospital discharge readiness in patients after myocardial infarction is at a similar level in three regions of Poland.

Although there are many publications on hospital discharge in the literature, they mainly concern children and parents of newborns after surgical procedures. There are few studies on hospital discharge readiness in patients with cardiovascular diseases, especially in patients after myocardial infarction.Perhaps the unsatisfactory resultsregarding patients’ high readiness for discharge from the hospital are caused by the lack of information in the questionnaire about caregiversupport and the lack of preparation of caregivers regarding the clinical condition of the patient who is discharged home. A qualitative descriptive study conducted by Melissa O Connor et al. showed that self-care ability, functional status, status of condition(s) and symptoms, the presence of a caregiver, support for the caregiver, connection to community resources/support, safety needs of the home environment, adherence to the prescribed regimen and care coordination all play an important role [17].

In Poland, in 2017, a program of coordinated specialist care for patients after a heart attack was launched by the ordinance of the Minister of Health. The program consisted of four modules: module I is “hospitalization”, which includes conservative treatment and invasive diagnostics; module II includes cardiac rehabilitation, which begins within 14 days of discharge from the hospital; module III is electrotherapy, under which the patient can, following an applicable sequence in the general mode, implement a cardioverter-defibrillator (ICD) or a cardiac resynchronization system (CRT-D); and module IVincludes an unlimited number of visits to a specialist clinic [18]. With reference to our research, which was conducted among patients after PCI surgery, preparation for discharge from the hospital took place as part of the first module, which provides important information on objective and subjective knowledge and expectations, and additional information on health, and the planning and development of further care during coordinated visits.

It is very difficult to compare our research results with other research results because there are few of them and they differ in terms of the research group.In this study, we found that half of the respondents (54.3%) were characterized by a low expectation of obtaining additional information. Expectations about information may be related to the expectation of some kind of support, in the form of information/knowledge.In a study by Wallace et al., whose purpose was to explore the association between patient characteristics and the patient- and nurse-completed Readiness for Hospital Discharge Scale, it was shown that patients’ perceived knowledge was lower in the case of people with insufficient reading and writing skills. Additionally, in the patient expectations subscale, a relationship was found between unmarried people and people living alone regarding the expectation of support after discharge [19]. In relation to our study, we found an additional characteristic that influenced patient expectations as a component of hospital discharge readiness: it was mostly male patients who had a higher need for more information about their disease and its treatment.The education of patients, carried out duringhospitalization after myocardial infarction, should be planned, monitored and conducted by all members of the therapeutic team in accordance with their competences [20]. Some studies have indicated that the continuity of nurse assignment during discharge care has the potential to increase patient readiness for discharge, which has been associated with fewer readmissions and emergency department visits [21]. Other research suggests the need for a structured process for discharging a patient from hospital, with tools to help healthcare organizations improve their discharge process and decrease readmission rates [22]. An additional tool may be the Chronic Disease Functioning Scale [14].

Our research showed a positive correlation between hospital discharge readiness and functioning with chronic disease in patients after myocardial infarction (r = 0.20; *p* < 0.001). It was shown that the higher the readiness for discharge from the hospital, the better the disease functioning and the lower the impact of the disease on the patient. It was also shown that the higher the level of subjective knowledge, the better the chronic disease functioning (rho = 0.16; *p* < 0.05), the greater the increase in the sense of influence over the course of the disease (rho = 0.17; *p* < 0.05) and the greater the decrease in the impact of the disease on the patient’s attitude (rho = 0.23, *p* < 0.05). In addition, the research showed that, with the increase in the level of objective knowledge, the level of disease functioning also increased (rho = 0.15; *p* < 0.05) and the disease’s impact on the patient decreased (rho = 0.17; *p* < 0.05).In addition to external resources, such as educating the patient about the disease, the ability to adapt to and cope with the situation related to myocardial infarctionmay affect their readiness for discharge from hospital. To better understand and assess the patient’s preparation for discharge from hospital, it is worth assessing the patient’s level of functioning with chronic disease [23].

Limitations of the Study

The limitations of this work include the small size of the study group (*n* =242) and the selection of patients from only one region in Poland (Silesian Region). Therefore, these results should not be usedto profile patients’ readiness for hospital discharge after MI worldwide.In addition, the limitations of this work include missing answers in the surveys and the long period of data collection. Therefore, care should be taken in assigning these results to the profile of all patients with MI, both nationwide and worldwide. This research needs to be extended in the future to expand the knowledge in this area; we plan to include more study groups in a multicenter study. Replicating this study with more data and more robust study designs is warranted to confirm these results.

## 5. Conclusions

The level of hospital discharge readiness in patients after MI is influenced by their level of functioning with chronic disease. The higher the readiness for discharge from the hospital, the better the disease functioning and the lower the impact of the disease on the patient. The assessment of readiness for discharge from the hospital should be guided not only by a questionnaire on readiness for discharge, but also by a comprehensive assessment of the social environment, support the patient receives and their level of functioning with a chronic illness. The main principle of proper patient preparation for hospital discharge should be an individual and holistic (biological, psychological and social) approach after myocardial infarction to prevent unplanned rehospitalization.

## Figures and Tables

**Figure 1 ijerph-20-01582-f001:**
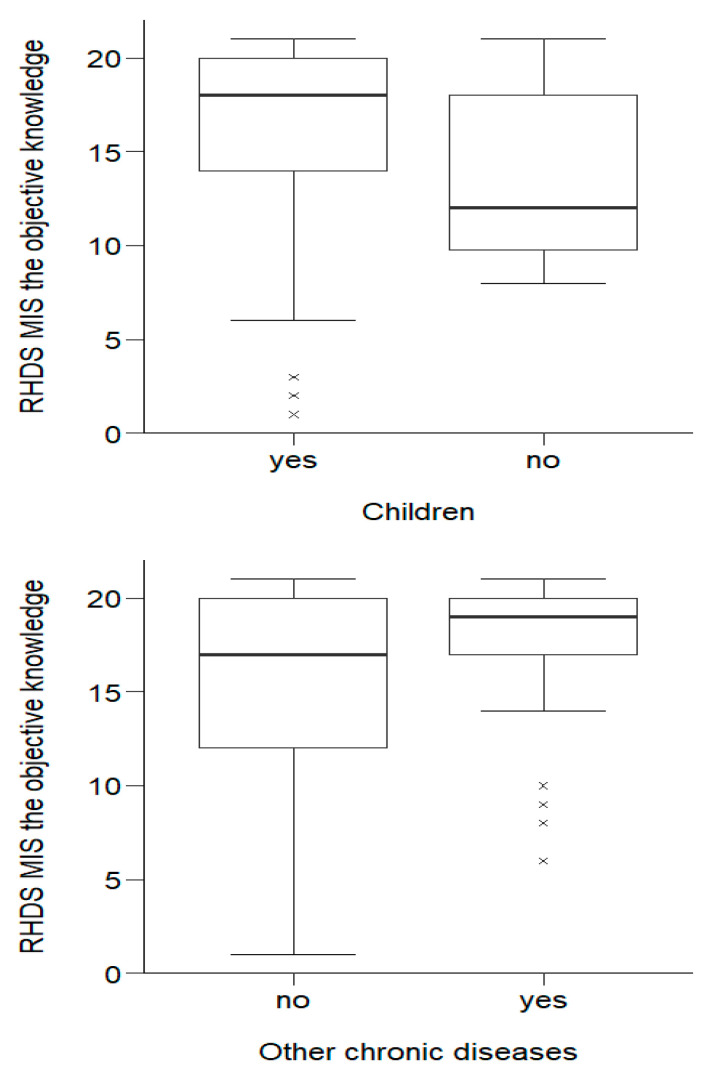
Effects of socio-demographic and clinical variables on objective knowledge about the disease in MI patients.

**Figure 2 ijerph-20-01582-f002:**
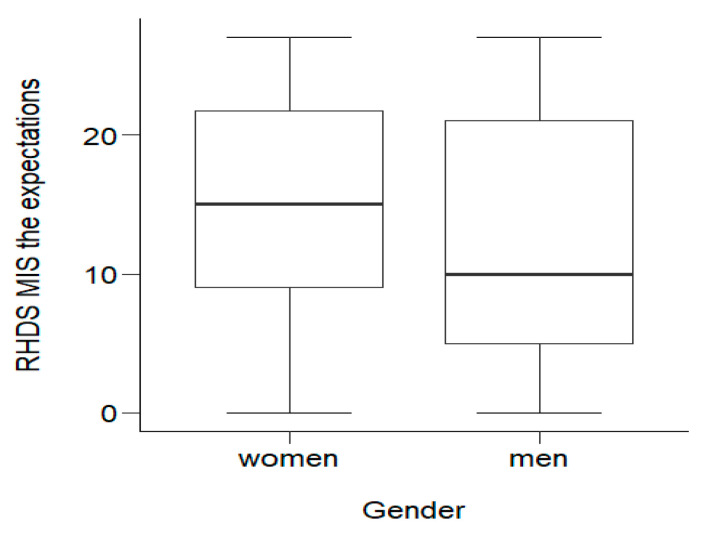
Effects of socio-demographic variables on MI patients’ expectationsof the disease.

**Figure 3 ijerph-20-01582-f003:**
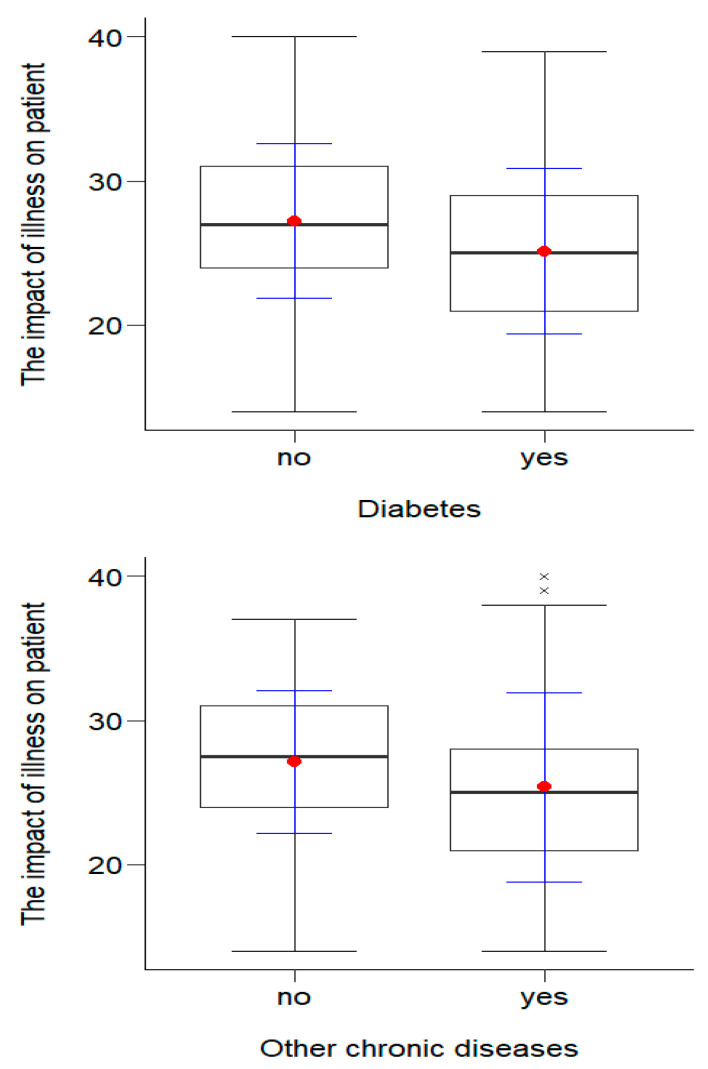
Effects of clinical variables on the impact of illness on the patients.

**Table 1 ijerph-20-01582-t001:** The socio-demographic and clinical variables on general readiness for hospital discharge in MI patients.

Variable		*n*	*M ± SD*	*Me*	*Q1–Q3*	*p*
Gender	women	70	48.16 ± 8.97	49.00	43.00–54.00	0.057 ^2^
	men	149	45.94 ± 11.37	45.00	39.00–53.00	
Age range	≤60 years	78	47.83 ± 12,06	47.50	39.00–57.00	0.392 ^4^
	61–70 years	73	47.47 ± 11.78	45.00	40.00–55.00	
	≥70 years	66	44.44 ± 7.05	45.50	39.00–50.00	
BMI range	normal	54	47.72 ± 10.19	48.00	40.00–54.00	0.959 ^3^
	overweight	93	46.65 ± 10.76	45.00	40.00–54.00	
	obesity	71	46.00 ± 11.05	45.00	39.00–53.00	
Marital status	in relationship	148	46.82 ± 11.26	46.00	39.00–54.00	0.917 ^1^
	single	57	46.65 ± 9.19	46.00	40.00–52.00	
Place of residence	city	107	47.17 ± 10.05	47.00	41.00–54.00	0.482 ^1^
	village	77	46.04 ± 11.59	45.00	39.00–54.00	
Education	primary	35	45.37 ± 9.78	45.00	40.00–53.00	0.681 ^3^
	vocational	88	46.11 ± 11.42	45.50	39.00–53.50	
	secondary	68	47.79 ± 9.82	48.00	41.50–54.00	
	higher	23	47.43 ± 12.10	45.00	39.00–57.00	
Professional activity	active	67	48.75 ± 11.74	48.00	39.00–58.00	0.176 ^2^
	inactive	100	45.70 ± 11.01	44.50	39.00–53.00	
Children	yes	184	46.87 ± 10.93	46.00	39.50–54.00	0.861 ^1^
	no	16	46.38 ± 9.07	44.50	41.00–49.50	
Arterial hypertension	no	71	47.63 ± 11.51	47.00	40.00–55.00	0.346 ^1^
	yes	148	46.18 ± 10.29	46.00	39.00–53.00	
Diabetes	no	152	47.19 ± 10.78	46.50	40.00–54.00	0.259 ^1^
	yes	67	45.42 ± 10.49	46.00	39.00–52.00	
Lipid disorders	no	164	46.48 ± 11.18	45.50	39.00–54.00	0.681 ^1^
	yes	55	47.16 ± 9.16	47.00	40.00–52.00	
Other chronic diseases	no	149	46.79 ± 11.10	47.00	39.00–54.00	0.783 ^1^
	yes	70	46.36 ± 9.84	45.00	40.00–53.00	
Smoking	currently	60	48.10 ± 10.30	47.50	42.50–54.50	0.307 ^3^
	never	105	47.19 ± 11.50	47.00	40.00–54.00	
	in the past	54	43.98 ± 9.09	42.00	37.00–48.00	
Myocardial infarction	first	178	46.80 ± 10.44	46.00	40.00–54.00	0.689 ^1^
	next	40	46.05 ± 12.00	44.50	37.00–53.50	
Total		219	46.65 ± 10.70	46.00	39.50–54.00	-
Readiness for hospital discharge	**Number of points**	**Interpretation**		** *n* **		** *%* **
	0–43	Low level		89		40.6%
	44–57	Intermediate level		94		42.9%
	58–69	High level		36		16.5%

*M*—mean; *SD*—standard deviation; Me—median; *Q1*—first quartile; *Q3*—third quartile; *p*—statistically significant results; ^1^*t*-test for independent trials; ^2^ U Mann–Whitney test; ^3^ one-way ANOVA; ^4^ test Kruskal–Wallis test; for each significant pair of response categories, the key of the category with the lower mean/median was recorded next to the category with the higher mean/median.

**Table 2 ijerph-20-01582-t002:** The socio-demographic and clinical variables on subjective knowledge of the disease in MI patients.

Variable		*n*	*M ± SD*	*Me*	*Q1–Q3*	*p*
Gender	women	70	17.03 ± 4.19	18.00	15.00–20.00	0.783 ^2^
	men	149	17.24 ± 4.61	18.00	14.00–21.00	
Age range	≤60 years	78	17.88 ± 4.79	19.00	15.00–21.00	0.232 ^4^
	61–70 years	73	16.88 ± 4.57	18.00	14.00–21.00	
	≥70 years	66	16.58 ± 3.93	17.00	15.00–20.00	
BMI range	normal	54	16.81 ± 4.22	17.00	15.00–21.00	0.786 ^4^
	overweight	93	17.54 ± 4.86	19.00	15.00–21.00	
	obesity	71	17.03 ± 4.15	18.00	14.00–21.00	
Marital status	in relationship	148	17.21 ± 4.59	18.00	15.00–21.00	0.376 ^2^
	single	57	17.61 ± 3.79	19.00	15.00–21.00	
Place of residence	City	107	17.36 ± 3.66	19.00	14.00–21.00	0.603 ^2^
	village	77	17.57 ± 3.72	18.00	15.00–21.00	
Education	primary	35	16.80 ± 4.34	17.00	15.00–20.00	0.403 ^4^
	vocational	88	16.93 ± 5.15	18.00	14.00–21.00	
	secondary	68	17.71 ± 3.56	18.50	15.00–21.00	
	higher	23	17.57 ± 3.89	19.00	16.00–21.00	
Professional activity	active	67	18.61 ± 4.55	19.00	17.00–21.00	0.293 ^2^
	inactive	100	17.41 ± 4.28	19.00	15.00–21.00	
Children	yes	184	17.36 ± 4.51	18.50	15.00–21.00	0.221 ^2^
	no	16	16.44 ± 3.16	15.00	14.50–20.00	
Arterial hypertension	no	71	17.82 ± 5.22	19.00	16.00–21.00	0.192 ^2^
	yes	148	16.86 ± 4.05	18.00	14.00–21.00	
Diabetes	no	152	17.39 ± 4.54	18.00	15.00–21.00	0.388 ^2^
	yes	67	16.69 ± 4.31	17.00	14.00–21.00	
Lipid disorders	no	164	17.24 ± 4.22	19.00	15.00–21.00	0.159 ^2^
	yes	55	16.98 ± 5.20	17.00	15.00–20.00	
Other chronic diseases	no	149	17.51 ± 4.30	18.00	15.00–21.00	0.311 ^2^
	yes	70	16.46 ± 4.78	18.00	14.00–21.00	
Smoking	currently	60	17.32 ± 5.50	18.00	15.00–21.00	0.182 ^4^
	never	105	17.35 ± 4.30	19.00	15.00–21.00	
	in the past	54	16.67 ± 3.45	17.00	14.00–20.00	
Myocardial infarction	first	178	17.35 ± 4.44	18.50	15.00–21.00	0.480 ^2^
	next	40	16.40 ± 4.62	17.00	14.00–21.00	
Total		219	17.17 ± 4.47	18.00	15.00–21.00	-
Subjective knowledge	**Number of points**	**Interpretation**		** *n* **		** *%* **
	0–15	Low level		73		33.3%
	16–18	Intermediate level		41		18.8%
	19–21	High level		105		47.9%

*M*—mean; *SD*—standard deviation; *Me*—median; *Q1*—first quartile; *Q3*—third quartile; *p*—statistically significant results; ^2^ U Mann–Whitney test; ^4^ test Kruskal–Wallis test; for each significant pair of response categories, the key of the category with the lower mean/median was recorded next to the category with the higher mean/median.

**Table 3 ijerph-20-01582-t003:** The socio-demographic and clinical variables on the Functioning in Chronic Illness Scale in MI patients.

Variable		*n*	*M ± SD*	*Me*	*Q1–Q3*	*p*
Gender	women	80	81.99 ± 11.73	82.00	73.50–88.50	0.347 ^1^
	men	162	83.52 ± 12.05	85.00	76.00–92.00	
Age range	≤60 years	85	85.04 ± 11.81	87.00	76.00–93.00	0.444 ^3^
	61–70 years	83	83.60 ± 11.48	85.00	77.00–90.00	
	≥70 years	72	79.93 ± 12.30	81.00	72.50–87.00	
BMI range	normal	57	83.09 ± 13.95	84.00	74.00–92.00	0.902 ^3^
	overweight	99	84.63 ± 11.22	85.00	79.00–93.00	
	obesity	84	81.06 ± 11.25	82.00	72.00–88.50	
Marital status	In relationship	162	83.41 ± 11.21	84.00	77.00–91.00	0.538 ^1^
	single	66	82.35 ± 13.24	84.00	74.00–91.00	
Place of residence	city	123	82.65 ± 11.14	84.00	76.00–89.00	0.383 ^1^
	village	83	84.08 ± 12.16	84.00	76.00–94.00	
Education	primary	40	82.63 ± 11.72	85.00	74.50–90.50	0.500 ^4^
	vocational	98	82.68 ± 11.90	82.50	76.00–90.00	
	secondary	74	82.43 ± 11.99	84.00	75.00–89.00	
	higher	25	87.04 ± 12.27	86.00	79.00–97.00	
Professional activity	active ^B^	74	86.04 ± 11.63	87.50	80.00–93.00	0.026 ^1^
	inactive	116	82.03 ± 12.18	82.00	74.00–90.00	
Children	yes	205	83.42 ± 11.42	85.00	76.00–91.00	0.065 ^1^
	no	17	77.94 ± 14.84	79.00	71.00–82.00	
Arterial hypertension	no^B^	83	85.37 ± 11.65	86.00	77.00–93.00	0.034 ^2^
	yes	159	81.79 ± 11.94	82.00	75.00–89.00	
Diabetes	no^B^	169	84.99 ± 11.41	86.00	80.00–92.00	<0.001 ^2^
	yes	73	78.45 ± 11.96	78.00	70.00–86.00	
Lipid disorders	no	184	83.76 ± 11.73	85.00	77.00–91.50	0.084 ^1^
	yes	58	80.66 ± 12.38	81.50	73.00–87.00	
Other chronic diseases	no ^B^	168	84.15 ± 10.99	85.00	78.00–91.00	0.040 ^1^
	yes	74	80.43 ± 13.60	81.00	71.00–89.00	
Smoking	currently	64	85.31 ± 11.56	87.00	78.00–93.00	0.191 ^3^
	never	114	82.21 ± 12.80	82.00	74.00–90.00	
	In the past	64	82.16 ± 10.52	84.00	76.50–88.00	
Myocardial infarction	first	199	83.64 ± 11.88	84.00	76.00–92.00	0.094 ^1^
	next	42	80.24 ± 12.06	85.00	72.00–89.00	
Total		242	83.02 ± 11.94	84.00	76.00–90.75	-
General functioning in the illness	**Number of points**	**Interpretation**		** *n* **		** *%* **
	24–78	Low level		76		31.4%
	79–93	Intermediate level		118		48.8%
	94–120	High level		48		19.8%

*M*—mean; *SD*—standard deviation; *Me*—median; *Q1*—first quartile; *Q3*—third quartile; *p*—statistically significant results; ^1^*t*-test for independent trials; ^2^ U Mann–Whitney test; ^3^ one-way ANOVA; ^4^ test Kruskal–Wallis test; ^B^ subsequent categories of answers; for each significant pair of response categories, the key of the category with the lower mean/median was recorded next to the category with the higher mean/median.

**Table 4 ijerph-20-01582-t004:** The socio-demographic and clinical variables regarding the patient’s impact on the illness.

Variable		*n*	*M ± SD*	*Me*	*Q1–Q3*	*p*
Gender	women	80	26.18 ± 4.88	25.50	23.00–29.00	0.005 ^2^
	men ^A^	162	27.60 ± 4.42	28.00	25.00–31.00	
Age range	≤60 years ^C^	85	28.21 ± 3.86	28.00	25.00–32.00	<0.001 ^4^
	61–70 years ^C^	83	27.39 ± 4.53	28.00	25.00–30.00	
	≤70 years	72	25.40 ± 5.06	25.50	23.00–28.00	
BMI range	normal	57	26.63 ± 5.18	26.00	24.00–30.00	0.676 ^4^
	overweight	99	27.34 ± 4.57	28.00	25.00–30.00	
	obesity	84	27.17 ± 4.32	27.00	24.00–30.50	
Marital status	in relationship	162	27.35 ± 4.20	28.00	25.00–30.00	0.483 ^2^
	single	66	27.05 ± 5.35	27.00	24.00–31.00	
Place of residence	city	123	27.37 ± 4.44	28.00	25.00–30.00	0.525 ^2^
	village	83	27.07 ± 4.85	27.00	24.00–30.00	
Education	primary	40	26.98 ± 4.05	26.00	24.00–30.00	0.123 ^4^
	vocational	98	26.70 ± 5.20	27.00	24.00–29.00	
	secondary	74	27.09 ± 3.73	28.00	25.00–29.00	
	higher	25	29.20 ± 4.84	31.00	25.00–33.00	
Professional activity	active ^B^	74	28.70 ± 3.75	29.00	26.00–32.00	0.020 ^1^
	inactive	116	27.30 ± 4.19	28.00	24.00–30.00	
Children	yes	205	27.42 ± 4.27	28.00	25.00–30.00	0.052 ^2^
	no	17	24.76 ± 6.76	25.00	23.00–29.00	
Arterial hypertension	no^B^	83	28.06 ± 3.72	28.00	25.00–31.00	0.034 ^2^
	yes	159	26.64 ± 4.96	27.00	24.00–29.00	
Diabetes	no^B^	169	27.55 ± 4.36	28.00	25.00–31.00	0.035 ^2^
	yes	73	26.15 ± 5.04	26.00	24.00–29.00	
Lipid disorders	no	184	27.48 ± 4.03	28.00	25.00–30.00	0.070 ^2^
	yes	58	26.00 ± 6.01	25.00	24.00–29.00	
Other chronic diseases	no	168	27.35 ± 4.64	28.00	25.00–31.00	0.167 ^2^
	yes	74	26.64 ± 4.56	27.00	24.00–30.00	
Smoking	currently	64	28.33 ± 3.82	28.00	25.00–32.00	0.060 ^4^
	never	114	26.65 ± 4.69	27.00	24.00–29.00	
	in the past	64	26.78 ± 5.04	27.00	24.50–30.00	
Myocardial infarction	first	199	27.32 ± 4.66	27.00	25.00–30.00	0.184 ^2^
	next	42	26.24 ± 4.41	26.50	24.00–29.00	
Total		242	27.13 ± 4.61	27.00	25.00–30.00	-
The patient’s impact on the illness	**Number of points**	**Interpretation**		** *n* **		** *%* **
	8–24	Low level		59		24.4%
	25–29	Intermediate level		115		47.5%
	30–40	High level		68		28.1%

*M*—mean; *SD*—standard deviation; *Me*—median; *Q1*—first quartile; *Q3*—third quartile; *p*—statistically significant results; ^1^*t*-test for independent trials; ^2^ U Mann–Whitney test; ^3^ one-way ANOVA; ^4^ test Kruskal–Wallis test; ^A–C…^ subsequent categories of answers; for each significant pair of response categories, the key of the category with the lower mean/median was recorded next to the category with the higher mean/median.

**Table 5 ijerph-20-01582-t005:** The effect of socio-demographic and clinical variables on patient’s attitude.

Variable		*n*	*M ± SD*	*Me*	*Q1*–*Q3*	*p*
Gender	women	80	28.69 ± 4.90	28.00	26.00–31.50	0.079 ^2^
	men	162	29.60 ± 5.37	30.00	26.00–34.00	
Age range	≤60 years	85	29.96 ± 5.47	30.00	27.00–34.00	0.802 ^3^
	61–70 years	83	29.53 ± 5.37	30.00	27.00–34.00	
	≥70 years	72	28.29 ± 4.71	28.00	25.00–32.00	
BMI range	normal	57	29.32 ± 5.99	29.00	25.00–34.00	0.736 ^3^
	overweight	99	29.94 ± 4.62	30.00	28.00–33.00	
	obesity	84	28.60 ± 5.35	28.00	25.00–33.00	
Marital status	in relationship	162	29.44 ± 4.95	30.00	27.00–33.00	0.301 ^2^
	single	66	28.77 ± 5.86	28.50	25.00–33.00	
Place of residence	city	123	28.83 ± 4.96	29.00	25.00–32.00	0.162 ^2^
	village	83	29.72 ± 5.31	30.00	27.00–33.00	
Education	primary	40	28.48 ± 5.06	29.00	25.00–32.00	0.821 ^4^
	vocational	98	29.31 ± 5.26	30.00	26.00–34.00	
	secondary	74	29.55 ± 5.38	29.00	27.00–32.00	
	higher	25	29.84 ± 5.00	30.00	27.00–33.00	
Professional activity	active ^B^	74	30.51 ± 5.37	31.00	28.00–35.00	0.017 ^2^
	inactive	116	28.73 ± 5.22	29.00	25.00–32.00	
Children	yes ^B^	205	29.40 ± 5.20	30.00	26.00–33.00	0.046 ^2^
	no	17	26.88 ± 5.60	27.00	23.00–30.00	
Arterial hypertension	no	83	30.07 ± 5.25	30.00	28.00–34.00	0.068 ^2^
	yes	159	28.89 ± 5.18	29.00	25.00–33.00	
Diabetes	no^B^	169	30.21 ± 4.98	30.00	28.00–34.00	<0.001 ^2^
	yes	73	27.18 ± 5.21	27.00	24.00–30.00	
Lipid disorders	no	184	29.43 ± 5.42	30.00	26.50–33.00	0.313 ^2^
	yes	58	28.88 ± 4.58	29.00	26.00–32.00	
Other chronic diseases	no	168	29.68 ± 5.00	30.00	27.00–33.00	0.147 ^2^
	yes	74	28.42 ± 5.65	29.00	25.00–33.00	
Smoking	currently	64	30.20 ± 5.23	30.50	27.00–34.00	0.397 ^3^
	never	114	28.99 ± 5.18	29.00	26.00–32.00	
	in the past	64	28.94 ± 5.29	29.50	25.00–32.50	
Myocardial infarction	first	199	29.48 ± 5.16	30.00	27.00–33.00	0.344 ^2^
	next	42	28.45 ± 5.59	29.00	25.00–32.00	
Total		242	29.30 ± 5.23	29.00	26.00–33.00	-
The impact of illness on patient’s attitude	**Number of points**	**Interpretation**		** *n* **		** *%* **
	8–27	Low level		78		32.2%
	28–33	Intermediate level		111		45.9%
	34–40	High level		53		21.9%

*M*—mean; *SD*—standard deviation; *Me*—median; *Q1*—first quartile; *Q3*—third quartile; *p*—statistically significant results; ^1^*t*-test for independent trials; ^2^ U Mann–Whitney test; ^3^ one-way ANOVA; ^4^ test Kruskal–Wallis test; ^B^ ubsequent categories of answers; for each significant pair of response categories, the key of the category with the lower mean/median was recorded next to the category with the higher mean/median.

## Data Availability

The data that support the findings of this study are available from the corresponding author, upon reasonable request.

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
