# Peer review of "The Variables of the Readiness for Discharge from Hospital in Patients after Myocardial Infarction"

_ijerph, 2023, doi:10.3390/ijerph20021582_

Round 1
Reviewer 1 Report
I would like to thank for being given the privilege of reviewing an interesting manuscript by Kolarczyk et al, on the perspective of readiness for discharge after myocardial infarction in a Polish cohort of patients. The authors should be congratulated for their important input in the field of a patient-centered approach, which should be one of the key focuses of any modern healthcare system.
Although I highly value the effort that the authors made to express their findings, I have several remarks. First, and most importantly, the authors used two scales, of which one (FCIS) even from the name, is clearly a risk score more suitable for long-term exposure to a chronic disease, and in the Reviewer’s opinion it would be much more appropriate e.g. during the rehabilitation programme after 6 weeks from MI. Moreover, I find it hard to determine what is the precise novelty of the manuscript, since readiness for discharge was evaluated even in the Polish cohorts as cited among the references in the manuscript.
With regard to the text itself, I have the following comments:
-The introduction is much too lengthy, I would suggest its shortening by at least 20-25%. Moreover, the Polish data indicate a stable, or even a lowering number of MIs in recent years, not an increase in the number as presented in the Text.
-The authors describe in the methods section that the study incorporated “a selected group of patients” however, it is unclear what was the selection based on. Is it just the exclusion of patients according to the criteria and other concomitant factors (“23 patients refused to participate in the study, and 36 respondents were not included in the statistical analysis due to missing answers in the sub-scales of the questionnaires”) or was it a different selection scheme? If so, could the selection bias influence the results?
I would recommend demonstrating on the illustration/graphic what was the exact number of patients, and who constituted the baseline assessment group.
-How was dementia assessed? Did the authors use any validated scale to check on dementia or any screening tool (including cognitive, behavioral, or daily-life assessment) tests?
-Characteristics of the study group are too lengthy, and some of the information is disposable, such as the long list of all comorbidities.
-I did not find the information on which day of hospitalization were the surveys collected. It is important since the patient’s perspective and education might be different on the first, than on the fifth day of hospital stay. Moreover, the stress associated with an acute phase of MI is
-The results of the subjective knowledge about the disease are rather interesting, especially since the degree of education or inhabitance location in some studies has been associated with increased knowledge of disease mechanisms and outcomes. Could the Authors comment more on that subject? Also, it is especially interesting that patients already after MI did not have any significantly higher knowledge of the disease than patients with 1st MI, and their readiness to be discharged was not different than that of patients after the 1st MI.
-In row 226, the authors claim that “Statistical analysis showed statistically significant (p=0.23) differences in the level of 226 expectations between men and women. “ although P=0.23 should be considered insignificant.
-Information on the Bonferroni test should be included in the Methods section prior to its presentation in Results.
-Are the Authors in the possession of the data, how many patients with MI (belonging to which group of readiness for discharge) agreed to participate in KOS-Zawał? The daily practice demonstrates a lower tendency for agreement to enroll at KOS in patients with lower subjective readiness to be discharged.
-It is interesting to note that (row 358) “it was mostly male patients who had a higher need for more information about their disease and its treatment” especially since subjectively, the majority of male patients are less likely to demand information on their health status. Could the Authors comment on that issue a bit more?
-Some more information about the limitations of the study (including scales validated solely in the Polish population) should be added.
-In the conclusions, the Authors state “Our findings suggest evidence for the useable of a hospital discharge readiness assessment questionnaire that may be helpful in qualifying a patient for coordinated care after myocardial infarction.” although little rationale for such statement is available in the Text. Could the authors elaborate on this subject more?
-Finally, there is plenty of spelling/grammar mistakes throughout the entire manuscript, which in my opinion would require elimination. Also, "Me" as an abbreviation for Median is not introduced in the text, and its determination was possible only owing to the Table.
Author Response
Dear Reviewer,
We thank you very much for the insightful review and interesting inquiries. Your remarks:
We thank you for pointing out the direction of new future research follow-up after 6 weeks rom MI in rehabilitation settings. We will consider such future research. However, this study was a cross sectional study design. The aim of the study was to determine whether and which variables, including the degree of functioning in chronic disease, affect the readiness to discharge a patient after myocardial infarction after undergoing PCI. Previous studies cited in the paper refer to readiness for hospital discharge in relation to adherence to treatment in patients after mocardial infraction (Kosobucka et al.) and acceptance of the patient's illness (Hydzik, et al.). Other studies have focused on factors of the discharge process and structural factors (Mabire at al.).The research was carried out in the cardiological rehabilitation ward and in the wards of university clinics. Our research involved another group of patients in interventional cardiology departments, which are a division of the private sector of medical care and it is dedicaded other variable like functionig in the chronic illness. However, the topic of hospital discharge readiness in patients after MI has not yet been fully explored, there are many dependencies and lack of knowledge in this area.
Your comments:
|
Reviewer comments: |
Author’s answers: |
|
|
1. |
-The introduction is much too lengthy, I would suggest its shortening by at least 20-25%. Moreover, the Polish data indicate a stable, or even a lowering number of MIs in recent years, not an increase in the number as presented in the Text. |
Thank you for your comment, we've shortened the introduction. However following the suggestion of the Reviewer, we removed information on epidemiological data. |
|
2. |
-The authors describe in the methods section that the study incorporated “a selected group of patients” however, it is unclear what was the selection based on. Is it just the exclusion of patients according to the criteria and other concomitant factors (“23 patients refused to participate in the study, and 36 respondents were not included in the statistical analysis due to missing answers in the sub-scales of the questionnaires”) or was it a different selection scheme? If so, could the selection bias influence the results? |
Thank you for your comment. There is definitely no selection bias. The study was conducted with strict inclusion and exclusion criteria described in the manuscript. The inclusion criteria for the study were: (1) adult participants diagnosed with myocardial infarction, as well as STEMI as NSTEMI(2) treated with PCI (Precutaneous Coronary Intervention), (3) informed consent to participate in the study, (4) with no dementia-related disorders and (5) with no mental disorders. Patients who did not consent to participate in the study, or who were unable to answer questions due to hearing disorders vision disorders, advanced senile dementia or diagnosed mental illnesses, were excluded from the studies. 23 patients refused to participate in the study, and 36 respondents were not included in the statistical analysis due to missing answers in the sub-scales of the questionnaires- this means that 23 people who refused to participate in the study and 36 people who skipped the entire subscales of the RHDS and FCIS questionnaire were not included in the data analysis (they were not included in the number N). We agree with the Reviewer's suggestion that this data presentation may be unclear, so we removed this information from the text. |
|
3. |
I would recommend demonstrating on the illustration/graphic what was the exact number of patients, and who constituted the baseline assessment group. |
Thank you for your suggestion. Graphics/illustration is generated by a statistical system in which the statistical analysis described in the methodology was performed. The Graphics has been done standard template generated by the program PSPP and we cannot modify it. A detailed description is placed in the text that refers to the figure. Therefore, the information contained in the text and in the illustration is not repeated and the graphics complement the text.
|
|
4. |
-How was dementia assessed? Did the authors use any validated scale to check on dementia or any screening tool (including cognitive, behavioral, or daily-life assessment) tests? |
Thank you for your comment. One of the criteria for exclusion from the study was the presence of mental disorders. This information was obtained from the patient'smedical record and form medical history. Patient was examinated by a physicians. The medical history wchich includes detailed in formation on comorbidities (including psychiatric, dementia). The patient's examination also used scales (NRS, VES-13, Norton) that were included in the medical records. If the patient's medical rekord contained information about mental illness and/or dementi- the patient was excluded from the study. |
|
5. |
-Characteristics of the study group are too lengthy, and some of the information is disposable, such as the long list of all comorbidities. |
Thank you for your comment, we have shortened the characteristics of the research group. |
|
6. |
-I did not find the information on chich day of hospitalization were the surveys collected. It is import ant since the patient’s perspective and education might be different on the first, than on the fifth day of hospitals tay. Moreover, the stress associated with an acute phase of MI is . |
Thank you for your comment. The surveys were collected on the last day of hospitalization. We have added this information to the text. The level of stress and coping with stress in patients in the acute phase of MI was not the aim of our study. According to the institution's data, in 2021 the average duration of hospitalization of patients was: 3.11 days and in 2022 - 3.03 days. The research question in our study was not how length of hospitalization affects hospital discharge readiness in patients with MI- we focused on other factors. |
|
7. |
The results of the subjective knowledge about the disease are rather interesting, especially since the degree of education or inhabitance location in some studies has been associated with increased knowledge of disease mechanisms and outcomes. Could the Authors comment more on that subject? Also, it is especially interesting that patients already after MI did not have any significantly higher knowledge of the disease than patients with 1st MI, and their readiness to be discharged was not different than that of patients after the 1st MI |
Thank you for your comment. Our study show that the subjective knowledge was slightly higher among men, people up to 60 years of age, overweight people, unmarried people, living in the countryside, with secondary education, professionally active. Slightly greater subjective knowledge was also observed among people with children, without comorbidities, not smoking at all, and people after the first heart attack. But the socio-demographic and clinical variables (the data) do not significantly differentiate the level of subjective knowledge in patients after myocardial infarction (p>0.005) - therefore the hypothesis was not confirmed. Following Reviewer’s comment we changed the presentation of these results in the text. |
|
8. |
-In row 226, the authors claim that “Statistical analysis showed statistically significant (p=0.23) differences in the level of 226 expectations between men and women. “ although P=0.23 should be considered insignificant. |
Thank you for your comment. One zero was missing when entering data into the text. The correct number is p=0.023 . This error has been corrected. |
|
9. |
-Information on the Bonferroni test should be included in the Methods section prior to its presentation in Results. |
Thank you for your comment. We added the Boniferroni test in the methodological section. |
|
10. |
-Are the Authors in the possession of the data, how many patients with MI (belonging to which group of readiness for discharge) agreed to participate in KOS-Zawał? The daily practice demonstrates a lower tendency for agreement to enroll at KOS in patients with lower subjective readiness to be discharged. |
Thank you for your comment. Daily practice from the center where the study was conducted shows 70% participation of All patients in the KOS-infarction program. The remaining 30% arepatients who refused to participate in the KOS-infraction program or had clinical contraindications, e.g. during dialysis therapy. Readiness for discharge from the hospital in patients after MI in relation to the KOS-infarction program was not the aim of our study. |
|
11. |
-It is interesting to note that (row 358) “it was mostly male patients who had a higher need for more information about their disease and its treatment” especially since subjectively, the majority of male patients are less likely to demand information on their health status. Could the Authors comment on that issue a bit more? |
Thank you for your comment. We've improved the text to make it clearer and more relevant. Please notice, that the study showed statistically insignificant (p> 0.05) differences in the level of subjective knowledge about the disease between women and men. Socio-demographic factors that significantly affect the level of subjective knowledge are described in detail on page 6, section 3.2.1. and in Table 2. |
|
12. |
-Some more information about the limitations of the study (including scales validated solely in the Polish population) should be added. |
Thank you for your comment. The research was carried out using two standardized measures: the Readiness for-Hospital Discharge After Myocardial Infarction Scale (RHD-MIS) and the Functioning in the Chronic Illness scale (FCIS). RHD-MIS is considered a reliable and relevant tool for measuring patient readiness for discharge. The a-Cronbach coefficient was 0.789 indicting a high level of reliability and homogeneity. The internal consistency of the questionnaire expressed by a-Cronbach coefficient was 0.855, indicates its high reliability and homogeneity. The use of a standardized survey questionnaire validated in the cultural population in which the research is conducted proves the methodological correctness of the research and is the strength of this research. Sharing research results by publishing in international journals in Open Acess gives greater access to free reading by other interested people from different countries and places in the world. RHD-MIS and FCIS are new scales whose validation was published in international journals in 2017 and 2018. Thiese study we study was interrupted and restarted due to the COVID pandemic. To date, there have been few studies using these scales. During the covid pandemic, many researchers focused on studying the pandemic, and other research, like ours, may have been suspended. Perhaps the publication of our research will interest and inspire other researchers from other countries who will read our research and also ask for the possibility of using this research tool. Please notice that in the limitations of the study contains the information about the selection of patients from only one region in Poland. However we added the information that: "it should be not taken in assigning these results to the profile at readiness for hospital discharge of patients after MI worldwide". |
|
13. |
-In the conclusions, the Authors state “Our findings suggest evidence for the useable of a hospital discharge readiness assessment questionnaire that may be helpful in qualifying a patient for coordinated care after myocardial infarction.” although little rationale for such statement is available in the Text. Could the authors elaborate on this subject more? |
Thank you for your comment. This issue is addressed in the discussion, however, we agree with the Reviewer that the conclusions concern the results of research and not of the discussion. Therefore, we have removed this provision from the conclusios. |
|
14. |
-Finally, there is plenty of spelling/grammar mistakes throughout the entire manuscript, which in my opinion would require elimination. Also, "Me" as an abbreviation for Median is not introduced in the text, and its determination was possible only owing to the Table. |
Thank you for your comment. We improved our manuscript. We addend the explanation of annabbreviation for Median in the text in „statistical analyses” section. We will made linguistic proofreading using Language Editing Services MDPI.
|
Once again, thank you for the review and we hope that our answers are satisfying for you.
Kind regards,
The authors

Reviewer 2 Report
The authors evaluated certain variables of the readiness for hospital discharge in patients with myocardial infarction (STEMI and NSTEMI) undergoing PCI. They used RHD-MIS and FCIS as validated scales for discharge readiness after MI and physical and mental functioning assessment, respectively. They found a positive correlation between hospital discharge readiness and functioning in chronic disease in myocardial infarction patients and a significantly better functioning in chronic disease. Also, there was an increase in the sense of influence on the course of the disease and a decrease in the impact of the disease on the patient's attitude with a higher level of subjective knowledge. The article is well- written, and it addresses an important aspect regarding the patients’ evolution after myocardial infarction. I would recommend for the “Characteristics of the Study Group” Section to also address the number (percent) of patients with STEMI and/or NSTEMI, localization of the MI and, if possible, left ventricular ejection fraction. Please carefully check and correct typos.
Author Response
Dear Reviewer,
we thank you very much for the insightful review and interesting inguiries.
Determination of NSTEMI and STEMI, localization of the MI and left ventricular ejection fraction is clinically important for prognosis. These data were not collected in our study because the aim of our study was to determine the variables that affect hospital discharge readiness, including the impact of functioning in a chronic disease – not prognosis. We will certainly consider the collection of these clinical characteristics in planning the future research on this topic. We will made linguistic proofreading using Language Editing Services MDPI.
Once again, thank you for the review and hope our answer is satisfactory for you.
Kind regards,
The authors
